# RAMM: Robust Adversarial Multimodal Learning for Protein Stability Prediction

## Abstract

Multimodal representations that integrate protein sequence and structure offer powerful priors for modeling protein properties, yet adapting them to small, task-specific datasets often leads to overfitting. We present RAMM, a two-stage adversarial multimodal learning framework for predicting the stability effects of protein mutations. In the first stage, we fine-tune a multimodal encoder on large protein datasets to capture general sequence–structure relationships. In the second stage, we train this encoder on target protein stability datasets while jointly optimizing an adversarial objective: a discriminator attempts to distinguish wild-type from mutant proteins, while the encoder learns to produce features robust to wildtype and mutation domain shifts, that fool the discriminator. This adversarial game drives the system toward a Nash equilibrium where the learned latent space becomes robust to distributional shifts introduced by mutations. Evaluations on low-sequence-identity benchmarks show that this approach improves generalization, achieving AUROC = 0.763 on the SKEMPI 2.0 classification task and RMSE = 1.39 kcal/mol on the S669 regression benchmark. These results highlight that adversarial deep learning can enhance the robustness and transferability of multimodal protein models for challenging biological prediction tasks.

## 1 Introduction

The computational prediction of protein function, stability, and interaction affinity is a cornerstone of modern molecular biology and therapeutic design. Progress has shifted from engineered descriptors to representation learning, where deep learning models automatically extract features from large-scale biological data. Current state-of-the-art methods remain largely unimodal: Protein Language Models (PLMs)( Weissenow & Rost (2025); Lin et al. (2023); Elnaggar et al. (2021)) leverage self-supervised learning on massive sequence databases to capture evolutionary context, while geometry-aware neural networks (Réau et al. (2023); Jha et al. (2022); Li et al. (2023); Zhao et al. (2023); Krapp et al. (2024)) explicitly model the 3D structural coordinates that dictate biophysical interactions. Yet these unimodal representations are incomplete, as a protein's biological identity is a composite of its 1D sequence, 3D relational graph, and global 3D geometry(Clark et al. (2019)).

Integrating such complementary modalities is thus a critical research frontier (Cheng et al. (2024); Xue et al. (2022); Hu et al. (2023); Nguyen & Hy (2024)). However, effective multimodal fusion is non-trivial due to the significant heterogeneity in the statistical properties and learned representations. Naive fusion strategies, such as the simple concatenation of feature vectors, often fail to adequately address this "heterogeneity gap", potentially leading to suboptimal representations where one modality dominates or where the synergistic potential of the combined data has not yet been fully realized(Guo et al. (2019)). This limitation necessitates the development of more principled fusion mechanisms capable of learning a unified representation that captures the shared, fundamental information across data modalities.

We propose a novel two-stage robust learning framework that separates general-purpose representation learning from adaptation to mutation stability prediction. Our hypothesis is that while a robust general representation can be learned through standard self-supervision, the learning process on smaller labeled datasets is prone to imbalance and may lose information that does not generalize across multiple modalities. We introduce RAMM, an adversarial learning algorithm applied during the supervised learning stage. The adversarial objective guides both classification and regression

training by integrating into the learning process, compelling the model to learn a representation optimized for mutation stability prediction while remaining robust. We evaluate our approach by performing unsupervised multimodal pre-training with our Fusion Autoencoder (FAE) on protein structures from the CATH database(Orengo et al. (1997); Sillitoe et al. (2020)).CATH organizes protein domains from the PDB into superfamilies based on ancestry, but in our setup, we ignore these labels and use only raw structures to learn task-agnostic features. This stage yields a powerful, general-purpose multimodal representation. We then demonstrate the superiority of RAMM by applying it to demanding downstream supervised tasks: predicting changes in protein stability upon mutation on the S669 benchmark(Pancotti et al. (2022); Zhang et al. (2023a)) and changes in protein–protein binding affinity on the SKEMPI 2.0 benchmark(Townshend et al. (2020); Jankauskaitė et al. (2019)). Our results show that adversarial regularization during learning achieves state-of-the-art performance in both classification and regression, establishing a robust and effective strategy for adapting large pre-trained multimodal models to specific biological problems.

In summary, our main contributions are:

- A self-supervised Fusion Autoencoder (FAE) that learns general-purpose multimodal representations from protein sequence and structure.
- RAMM, a two-stage adversarial learning framework that leverages the FAE to learn robust, generalizable features.
- RAMM achieves state-of-the-art performance on S669 (protein stability) and SKEMPI 2.0 (binding affinity) benchmarks in both classification and regression downstream tasks for mutation stability prediction

## 2 RELATED WORK

**Protein Representation Learning Paradigms**    The field of protein representation learning has been dominated by two distinct yet complementary paradigms. First, sequence-based Protein Language Models (PLMs) such as ESM-1(Rives et al. (2021)), ESM-2(Lin et al. (2023)), Protein-BERT(Brandes et al. (2022)) and ProtTrans(Elnaggar et al. (2021)) have achieved remarkable success by treating protein sequences as a biological language, training large transformer architectures on billions of sequences to capture deep evolutionary context(Zhang et al. (2023c)). Pre-trained via masked language modeling, these models have demonstrated strong performance across a wide range of downstream tasks. Notably, scaling ESM-2 to billions of parameters has led to emergent capabilities, including the direct prediction of three-dimensional protein structures from sequence at speeds far exceeding traditional pipelines. However, recent analyses indicate that PLMs excel primarily on tasks well-aligned with their pre-training objectives(Lin et al. (2023); Xu et al. (2022)), with their performance sometimes dropping as model size increases on other biological applications. This suggests that sequence-based pre-training alone may be insufficient to capture the full biophysical and functional complexity of proteins.

**Structure-Aware Protein Models**    In parallel, structure-aware approaches have emerged to explicitly model the three-dimensional geometric constraints that govern protein function. Graph Neural Networks (GNNs) such as GVP-GNN and other geometric deep learning methods(Jing et al. (2020; 2021); Hsu et al. (2022)) have shown great promise by incorporating 3D coordinates and spatial relationships into their architectures. Models like Orientation-Aware GNNs (OAGNNs)(Li et al. (2025)) have further refined this by enforcing SO(3)-equivariance to better capture fine-grained geometric features like torsion angles, leading to strong performance on tasks that demand detailed structural reasoning, including binding site prediction and model quality assessment(Deng et al. (2021). However, structure-based models face significant scalability challenges due to the relative scarcity of high-quality, experimentally resolved structures and the computational cost of processing irregular 3D geometries.

**Multimodal Fusion Approaches**    Given that sequence and structure provide complementary perspectives, integrating these modalities has become a critical research frontier(Kalifa et al. (2025); Zhang et al. (2023b)). Earlier approaches relied on simple feature concatenation, which often failed to capture meaningful cross-modal interactions. More advanced methods now employ attention-based mechanisms, such as the Bi-Hierarchical Fusion Framework(Liu et al. (2025)), which enables a continuous, bidirectional flow of information between sequence and structure encoders.

Contrastive alignment has also emerged as a powerful paradigm; tri-modal frameworks like ProteinAligner(Zhang et al. (2024) learn a shared embedding space that aligns sequence, structure, and textual descriptions from scientific literature. Other notable models, such as MPRL(Hu et al. (2023) and MESM(Wang et al. (2025), fuse sequence representations with residue-level graphs and 3D point clouds, using hybrid auto-encoder architectures to extract and combine multimodal features. These works establish the clear benefit of multimodal fusion but often leave open the question of how to best adapt these powerful, pre-trained representations to specific, data-limited downstream tasks.

**Protein Stability Prediction**   Predicting protein stability changes upon mutation ($\Delta\Delta G$) is a critical benchmark for evaluating protein representations(Gong et al. (2023); Cao et al. (2019)). Classical physics-based methods are accurate but computationally expensive, while recent deep learning models, such as ThermoMPNN(Dieckhaus et al. (2024)) and DDGemb(Savojardo et al. (2025)), achieve competitive performance through transfer learning and PLM embeddings. Yet, most models perform poorly on the challenging S669 benchmark, highlighting limited generalization to low-similarity proteins. Although multimodal strategies are emerging, state-of-the-art approaches remain largely unimodal, leaving the potential of multimodal fusion underexplored.

**Adversarial Domain Adaptation**   Adversarial learning has shown promise in addressing domain shifts across various biological settings, yet its application to protein representation learning remains nascent(Carbone et al. (2022); Lan et al. (2024); Yuan et al. (2023)). Methods such as AITL(Sharifi-Noghabi et al. (2020)) have successfully employed adversarial training to bridge the gap between preclinical and clinical datasets in pharmacogenomics, while graph-based adversarial techniques(Ünlü et al. (2025); Hadipour et al. (2025)) have been used to tackle cross-network transfer problems. Despite these successes, adversarial regularization has rarely been explored in the context of multimodal protein representation learning, particularly as a mechanism to improve generalization during fine-tuning. This gap is especially relevant for stability prediction, where training data are limited and often distributionally different from test proteins. While it is hypothesized that incorporating adversarial objectives could help improve out-of-distribution generalization(Shi et al. (2025)), this remains a critically under-explored direction, one that we address directly in this work.

## 3 Method

Our framework, RAMM, is designed as a two-stage learning pipeline that decouples general-purpose representation learning from downstream mutation prediction. The first stage involves self-supervised learning of a multimodal Fusion Autoencoder (FAE) on a large, unlabeled structural database to learn a robust, protein representation. In the second stage, RAMM applies an adversarial objective during supervised learning, regularizing the training process to enforce robustness and improve generalization across multiple modalities. Overall workflow is shown in Figure 1

### 3.1 Unimodal Feature Extraction

We begin by representing each protein through three complementary modalities: its 1D sequence, its 3D relational graph, and its 3D atomic point cloud. To ensure a high-quality feature space, we employ state-of-the-art, publicly available deep learning models as frozen feature extractors for each modality. These extractors are applied consistently to generate unimodal representations for all proteins in our pre-training (CATH) and downstream (SKEPMI 2.0, S669) datasets.

- Sequence Representation ($z_{\text{seq}}$): The primary amino acid sequence of a protein provides deep evolutionary context. We use the pre-trained Prot5-XL-U50 model(Elnaggar et al. (2021)), a 4.2-billion parameter Protein Language Model (PLM), to generate a fixed-size 1024-dimensional embedding for each protein sequence. This representation is obtained by taking the mean of the final hidden layer's outputs across all residue positions.
- Graph Representation ($z_{\text{graph}}$): To capture the relational structure and local chemical environment, we model the protein as a graph $\mathcal{G} = (\mathcal{V}, \mathcal{E})$, where each node $v \in \mathcal{V}$ corresponds to an amino acid residue. Edges $e \in \mathcal{E}$ are constructed between the alpha-carbon ($C_\alpha$) atoms of the five nearest neighboring residues in 3D Euclidean space. This graph is then processed by a pre-trained E(n) Equivariant Graph Neural Network (EGNN)(Satorras et al.

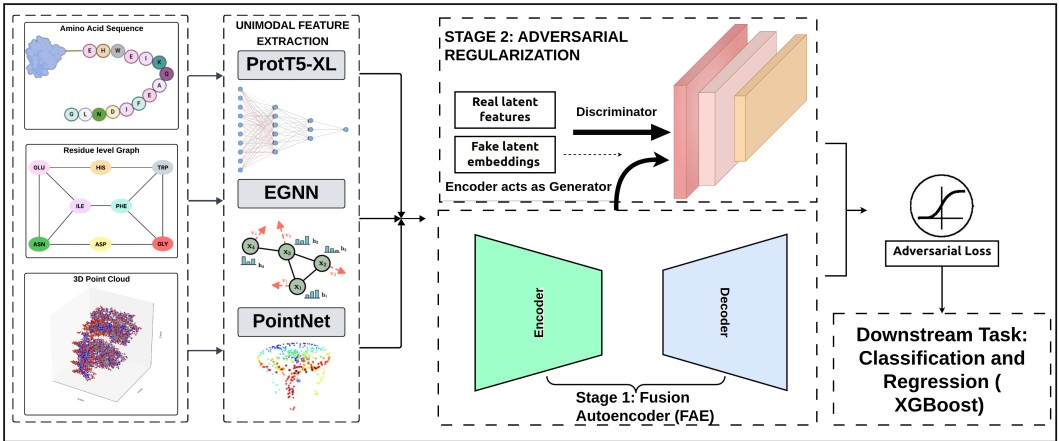

Figure 1: Overview of our RAMM framewrok

(2021)), which generates a single graph-level embedding that is invariant to the protein's rotation and translation in 3D space.

- Point Cloud Representation ($z_\text{pcloud}$): To encode the protein's global geometry and shape, we represent its atomic coordinates as a 3D point cloud(de Villiers et al. (2023)). Prior to feature extraction, the point cloud undergoes a standardization process to ensure uniformity across different protein structures: (i) it is centered by translating its centroid to the origin $(0, 0, 0)$; (ii) it is scaled to fit within a unit sphere; and (iii) it is sampled or padded to a fixed size of 2048 points. A pre-trained PointNet model then processes this standardized point cloud to extract a single global feature vector $z_\text{pcloud}$, which is invariant to the permutation of the input atoms.

### 3.2 STAGE 1: SELF-SUPERVISED PRE-TRAINING FOR MULTIMODAL FUSION

The objective of the learning stage is to learn a general-purpose fusion of the three unimodal representations. We achieve this using the Fusion Autoencoder (FAE), trained on the large-scale CATH structural database.

The input to our FAE is the concatenation of the unimodal feature vectors,

$$\mathbf{z}_\text{in} = [\mathbf{z}_\text{seq} \,\|\, \mathbf{z}_\text{graph} \,\|\, \mathbf{z}_\text{pcloud}]. \tag{1}$$

Prior to concatenation, each modality is projected into a common dimensional space using a projection head (Linear $\rightarrow$ ReLU $\rightarrow$ LayerNorm).

The FAE consists of a multi-layer perceptron (MLP) encoder $E_\phi$ and a corresponding MLP decoder $D_\psi$, parameterized by $\phi$ and $\psi$ respectively. The encoder maps the concatenated input to a compressed latent representation $\boldsymbol{h}$:

$$\boldsymbol{h} = E_\phi(\mathbf{z}_\text{in}). \tag{2}$$

The encoder consists of two fully connected layers with Tanh and ReLU activations, while the decoder mirrors this structure with fully connected layers and ReLU.

The decoder then attempts to reconstruct the original concatenated vector from this latent representation:

$$\hat{\mathbf{z}}_\text{in} = D_\psi(\boldsymbol{h}) \tag{3}$$

The model is trained in a self-supervised manner by minimizing the Mean Squared Error (MSE) reconstruction loss, $\mathcal{L}_\text{recon}$, between the original and reconstructed vectors:

$$\mathcal{L}_\text{recon} = \frac{1}{N} \sum_{i=1}^{N} \left\| \mathbf{z}_\text{in}^{(i)} - \hat{\mathbf{z}}_\text{in}^{(i)} \right\|_2^2 \tag{4}$$

This process forces the encoder $E_\phi$ to learn a dense latent space that captures the essential, shared information across all three modalities.

## 3.3 STAGE 2: ADVERSARIAL REGULARIZATION FOR SUPERVISED FINE-TUNING

After the initial learning, the encoder $E_\phi$ serves as a feature extractor. In second stage, RAMM trains this encoder to handle distributional shifts introduced by protein mutation using a self-supervised, adversarial adaptation objective. or downstream tasks, we use the trained encoder to generate latent representations for wild-type ($h_{\mathrm{wt}}$) and mutant ($h_{\mathrm{mut}}$) proteins. We then derive the difference vector

$$\Delta h = h_{\mathrm{mut}} - h_{\mathrm{wt}} \tag{5}$$

which captures the effect of the mutation in latent space, and use it as input features for a supervised predictor.

For downstream tasks, we use the final, trained encoder to generate latent representations for wild-type ($h_{\mathrm{wt}}$) and mutant ($h_{\mathrm{mut}}$) proteins. The difference vectors (5) are then used as input features for a supervised predictor. The purpose of the adversarial stage is to regularize the latent space, making it more robust for differencing. We reframe the problem as one of domain adaptation, with wild-type proteins as the *source* domain and mutant proteins as the *target* domain. The objective is to learn a representation space where both domains are distributionally aligned.

This is achieved by jointly training a set of three modality-specific discriminators, $\{D_{\omega_{\mathrm{seq}}}, D_{\omega_{\mathrm{graph}}}, D_{\omega_{\mathrm{pcloud}}}\}$, alongside the encoder. Each discriminator is implemented as an MLP and operates on the *final fused latent representation $h$*, rather than intermediate unimodal features. This design forces the entire fusion process to become domain-invariant. For training we used $\mathrm{LR}_{\mathrm{enc}} = 1 \times 10^{-5}$ for the encoder and $\mathrm{LR}_{\mathrm{disc}} = 5 \times 10^{-5}$ for discriminator. Each modality was projected into a 640 dimensional space and fused into a 1024 dimensional latent vector. The reconstruction loss was weighted by 1.0, and the adversarial weights $[0.2, 0.1, 0.1]$.

The training proceeds as a two-player minimax game between the fusion encoder $E_\phi$ (the "generator") and the set of discriminators $D_\omega$. For each modality-specific discriminator $D_{\omega_m}$, the objective is to find a Nash equilibrium of the following value function:

$$\min_{E_\phi} \max_{D_{\omega_m}} V(D_{\omega_m}, E_\phi) = \mathbb{E}_{h_{\mathrm{mut}} \sim P_{\mathrm{mut}}} \big[ \log D_{\omega_m}(h_{\mathrm{mut}}) \big]$$
$$+ \mathbb{E}_{h_{\mathrm{wt}} \sim P_{\mathrm{wt}}} \big[ \log \big( 1 - D_{\omega_m}(E_\phi(Z_{\mathrm{in,wt}})) \big) \big]. \tag{6}$$

In practice, this minimax game is optimized by alternating between two updates:

**Maximizing $V$ w.r.t. $D$ (Discriminator Update).** For a fixed encoder, each discriminator $D_{\omega_m}$ is trained to distinguish latent vectors from mutant proteins ("real") and wild-type proteins ("fake"). This corresponds to ascending the stochastic gradient of

$$\mathcal{L}_{\mathrm{adv},D,m} = \mathbb{E}_{h_{\mathrm{mut}}} \big[ \log D_{\omega_m}(h_{\mathrm{mut}}) \big] + \mathbb{E}_{h_{\mathrm{wt}}} \big[ \log \big( 1 - D_{\omega_m}(h_{\mathrm{wt}}) \big) \big]. \tag{7}$$

**Minimizing $V$ w.r.t. $E$ (Encoder Update).** For fixed discriminators, the encoder $E_\phi$ is trained to produce wild-type embeddings that fool the discriminators. This corresponds to descending the stochastic gradient of

$$\mathcal{L}_{\mathrm{adv},G,m} = -\mathbb{E}_{h_{\mathrm{wt}}} \big[ \log D_{\omega_m}(h_{\mathrm{wt}}) \big]. \tag{8}$$

**Total Encoder Loss.** This adversarial process is integrated with the self-supervised reconstruction objective to stabilize training and prevent catastrophic forgetting. The encoder is updated using

$$\mathcal{L}_{\mathrm{total\text{-}enc}} = \mathcal{L}_{\mathrm{recon}} + \sum_{m \in \{\mathrm{seq},\mathrm{graph},\mathrm{pcloud}\}} \lambda_m \, \mathcal{L}_{\mathrm{adv},G,m}, \tag{9}$$

where $\mathcal{L}_{\mathrm{recon}}$ is the MSE reconstruction loss applied to both WT and MUT samples, and $\lambda_m$ are modality-specific weights.

This process regularizes the encoder, compelling it to learn a common, mutation-aware latent space that aligns WT and MUT domains while retaining structural information essential for downstream mutation effect prediction.

---

**Algorithm 1** Two-Stage Training: Fusion Autoencoder Pretraining and Adversarial Fine-Tuning

**Stage 1: Self-Supervised Pretraining of Fusion Autoencoder (FAE)**
**Require:** Unimodal inputs($\mathbf{z}_{\text{seq}}, \mathbf{z}_{\text{graph}}, \mathbf{z}_{\text{pcloud}}$), encoder $E_\phi$, decoder $D_\psi$ from CATH dataset
 1: **for** each minibatch **do**
 2:     Forward pass through FAE: $\hat{\mathbf{z}}_{\text{in}} = D_\psi(E_\phi(\mathbf{z}_{\text{in}}))$(1)(3)(2)
 3:     Update parameters $\phi, \psi$ by minimizing reconstruction loss $\mathcal{L}_{\text{recon}}$ (4)
 4: **end for**
    **Stage 2: Adversarial Fine-Tuning of Mutation-Aware Encoder**
**Require:** Wild-type inputs $\mathbf{z}_{\text{in,wt}}$, mutant inputs $\mathbf{z}_{\text{in,mut}}$; pretrained encoder $E_\phi$; discriminators $D_{\omega_{\text{seq}}}$,
    $D_{\omega_{\text{graph}}}, D_{\omega_{\text{pcloud}}}$ using SKEMPI and S669 dataset
 5: **for** each minibatch **do**
 6:     Encode wild-type and mutant: $\mathbf{h}_{\text{wt}} = E_\phi(\mathbf{z}_{\text{in,wt}})$, $\mathbf{h}_{\text{mut}} = E_\phi(\mathbf{z}_{\text{in,mut}})$
 7:     Compute discriminator loss for 3 discriminators $\mathcal{L}_{\text{adv},D}$ (7)
 8:     Ascend $\nabla_\omega \mathcal{L}_{\text{adv},D}$
 9:     Compute reconstruction loss $\mathcal{L}_{\text{recon}}$
10:     Compute adversarial generator loss for 3 modalities (8)
11:     Form total encoder loss: $\mathcal{L}_{\text{total-enc}} = \mathcal{L}_{\text{recon}} + \sum_m \lambda_m \mathcal{L}_{\text{adv},G,m}$ (9)
12:     Descend $\nabla_\phi \mathcal{L}_{\text{total-enc}}$
13: **end for**
14: Obtain latent embeddings $\mathbf{h}_{\text{wt}}, \mathbf{h}_{\text{mut}}$ with trained encoder
15: Compute mutation effect vector $\Delta\mathbf{h} = \mathbf{h}_{\text{mut}} - \mathbf{h}_{\text{wt}}$
16: Train supervised predictor on $\Delta\mathbf{h}$

---

# 4 EXPERIMENTAL SETUP

## 4.1 DATASETS

Our experimental design leverages a large, structurally diverse dataset for self-supervised pre-training and three distinct, widely recognized benchmarks for downstream task evaluation.

**Pre-training Dataset.** We use the CATH v4.3(Orengo et al. (1997)) database for the initial self-supervised learning of our Fusion Autoencoder. CATH is a comprehensive, hierarchical classification of protein domains, providing a large and structurally diverse set of proteins ideal for learning general-purpose representations. We utilize the protein sequences and their corresponding 3D structures from this database, deliberately ignoring the CATH classification labels to ensure our pre-training is entirely self-supervised.

## 4.2 PERFORMANCE ON LOW-IDENTITY MUTATION STABILITY CLASSIFICATION

To test our central hypothesis, that adversarial regularization during fine-tuning improves generalization to unseen protein families, we evaluate our framework on the Mutation Stability Prediction (MSP) benchmark.

### 4.2.1 DATASET

We evaluate on the MSP benchmark, derived from SKEMPI 2.0 and curated as part of Atom3D(Townshend et al. (2020); Jankauskaitė et al. (2019)). It contains 4,148 mutant structures paired with 316 wild-type complexes, annotated by changes in binding free energy ($\Delta\Delta G$). We treat this as a binary classification task: mutations improving binding energy are labeled stabilizing (1), and others destabilizing (0). A strict 30% sequence identity cutoff between training and testing sets ensures generalization to unseen protein folds.

### 4.2.2 SETUP

We generated latent embeddings for each wild-type ($\boldsymbol{h}_{wt}$) and mutant ($\boldsymbol{h}_{mut}$) structure using our pretrained and adversarially fine-tuned multimodal encoder. Following the methodology outlined in Section 3.3, mutation-level representations were computed as difference vectors ($\Delta\boldsymbol{h} =$

$h_{mut} - h_{wt}$), which explicitly capture the semantic shift in the latent space induced by the mutation. An XGBoost classifier was trained on these difference vectors. To account for the natural class imbalance (destabilizing mutations are more common), we evaluated two settings: (i) standard training with class-weighted loss and (ii) training on a SMOTE-augmented version of the training data. Models were trained on 80% of the data and evaluated on a held-out 20% test split.

### 4.2.3 BASELINES

We compared our framework against a comprehensive set of representative baselines reported on this benchmark, spanning sequence-based, structure-based, and multimodal paradigms. These include ESM-2(Lin et al. (2023)), GNN(Townshend et al. (2020)), GVP(Jing et al. (2020)), ESM-GearNetZhang et al. (2023c)), MPRL(Nguyen & Hy (2024)), and FusionProt(Kalifa et al. (2025)).

### 4.2.4 RESULTS

Our approach substantially outperforms established baselines (Table 1), demonstrating that adversarially tuned representations generalize well. The **class-weighted XGBoost** model (Table 2)achieves an AUROC of 0.763. At the default threshold of 0.5, it attains **80.5% accuracy**, with a balanced accuracy of 0.609 and an MCC of 0.308. On a per-class basis, precision and recall are 0.825 and 0.954 for the stabilizing (majority) class, and 0.610 and 0.264 for the destabilizing (minority) class, reflecting strong sensitivity to the majority class.

Table 1: AUROC performance of different sequence, structure, and multimodal baselines on the MSP benchmark. Our adversarially fine-tuned model substantially outperforms previous methods.

| Approach | AUROC |
|---|---|
| *Sequence-only Baselines* | |
| ESM-2(Lin et al. (2023) | 0.509 |
| Rao et al.(Rao et al. (2019) | 0.554 |
| *Structure-only Baselines* | |
| 3DCNN(Townshend et al. (2020)) | 0.574 |
| ENN(Townshend et al. (2020)) | 0.574 |
| GNN(Townshend et al. (2020)) | 0.609 |
| VGAE(Nguyen & Hy (2024)) | 0.463 |
| PAE(Nguyen & Hy (2024)) | 0.488 |
| GVP(Jing et al. (2021)) | 0.709 |
| *Multimodal Baselines* | |
| ESM-GearNet(Zhang et al. (2023c)) | 0.599 |
| MPRL(Nguyen & Hy (2024)) | 0.612 |
| FusionProt(Kalifa et al. (2025)) | 0.745 |
| **RAMM (ours)** | **0.763** |

Table 2: Performance of the class-weighted XGBoost model on the MSP benchmark under the default (0.5) and optimized (0.261) thresholds. AUROC is threshold-independent, while the optimized threshold maximizes F1.

| Metric | Default (0.5) | Optimized (0.261) |
|---|---|---|
| Accuracy | 0.805 | 0.772 |
| AUROC | 0.763 | 0.763 |
| Balanced Accuracy | 0.609 | **0.692** |
| MCC | 0.308 | **0.365** |
| Precision (Destabilizing) | 0.610 | 0.476 |
| Recall (Destabilizing) | 0.264 | **0.551** |

To better reflect performance under imbalance, we optimized the decision threshold for maximum F1 (0.261). At this threshold, the model attains a balanced accuracy of 0.692 and an MCC of 0.365, with recall for destabilizing mutations improving to 0.551 and precision for that class of 0.476. The ROC curves (Figure 2) further confirm stable discriminative power across the full operating range.

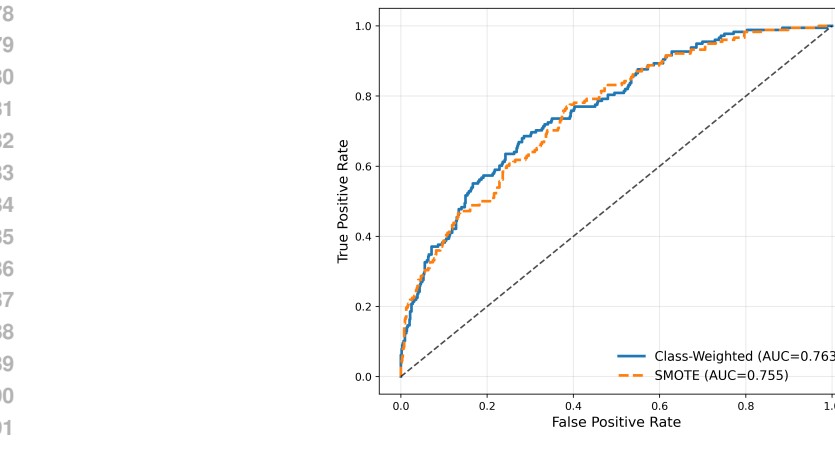

Figure 2: ROC curve comparison of class-weighted and SMOTE-augmented XGBoost models on the MSP benchmark. The class-weighted model achieves slightly higher AUROC, indicating better class separation without synthetic oversampling.

The slightly higher AUROC for the class-weighted model compared to SMOTE (0.763 vs. 0.755) suggests that weighting preserves the natural class distribution and maintains cleaner class separation, while SMOTE improves minority recall but at the cost of introducing synthetic overlap between classes.

### 4.3 PERFORMANCE ON PROTEIN STABILITY REGRESSION

To complement our classification results, we performed a rigorous evaluation of our framework on the task of protein stability regression, where the goal is to predict the continuous change in Gibbs free energy ($\Delta\Delta G$).

#### 4.3.1 DATASET AND SETUP

We benchmarked our regression approach on the S669 dataset(Pancotti et al. (2022); Zhang et al. (2023a)), containing 669 single-point mutations with experimentally measured $\Delta\Delta G$ values and $< 25\%$ sequence identity to other databases, posing an out-of-distribution challenge.

Using difference vectors ($\Delta\boldsymbol{h} = \boldsymbol{h}_{mut} - \boldsymbol{h}_{wt}$) from our adversarially fine-tuned encoder as features, we trained an XGBoost model to predict $\Delta\Delta G$, with hyperparameters optimized via grid search and 5-fold cross-validation.

#### 4.3.2 BASELINES

We compare our results against state-of-the-art methods previously evaluated on S669. The baselines span a wide range of paradigms, including modern sequence-based predictors that leverage large language models (e.g., DDGemb(Savojardo et al. (2025)), PROSTATA(Umerenkov et al. (2023)), ThermoMPNN(Dieckhaus et al. (2024)) and established structure-based approaches (e.g., ACDC-NN(Benevenuta et al. (2021))), providing a robust context for our performance.

#### 4.3.3 RESULTS

Our adversarially tuned multimodal representation demonstrates strong predictive accuracy, achieving the lowest error rates among all compared methods. The results are summarized in Table 3. The results highlight a key finding: while some large sequence-only models like DDGemb achieve higher Pearson correlation, our model attains a lower RMSE of 1.39 kcal/mol and a state-of-the-art MAE of 0.99 kcal/mol, indicating predictions closer to experimental values. This supports our hypothesis that sequence-only models, despite strong correlations from large pre-training, can incur larger errors on out-of-distribution samples like S669. In contrast, our adversarially regularized

Table 3: Comparison of predictive performance on $\Delta\Delta G$ estimation across sequence, structure, and multimodal methods. PCC: Pearson correlation coefficient (higher is better), RMSE: root mean squared error (lower is better), MAE: mean absolute error (lower is better).

| Method | Modality | PCC ↑ | RMSE ↓ | MAE ↓ |
|---|---|---|---|---|
| DDGemb | Sequence | 0.68 | 1.40 | 0.99 |
| PROSTATA | Sequence | 0.65 | 1.45 | 1.00 |
| ACDC-NN | Structure | 0.61 | 1.50 | 1.05 |
| ThermoMPNN | Sequence | 0.43 | 1.52 | - |
| **RAMM (ours)** | Multimodal | 0.52 | 1.39 | 0.99 |

multimodal framework, integrating structural and geometric information, produces more physically accurate and robust predictions.

## 4.4 ABLATION STUDY

To quantify the impact of each input modality in our fused representation, we conducted a controlled ablation in which ProtT5, PointNet, and EGNN features were removed one at a time while keeping the full training pipeline unchanged. The fusion module, adversarial fine-tuning, optimization settings, and downstream predictor were kept identical across all variants; only the ablated modality was excluded from the input. This allows us to assess how much each representation contributes to the mutation-sensitive latent space.

**SKEMPI (Mutation Stability Classification).** Table 4 summarizes the results. Removing any modality reduces performance relative to the full model. ProtT5 ablation yields the largest AUROC drop ($0.763 \rightarrow 0.704$), reflecting the importance of evolutionary signal. Removing PointNet lowers AUROC and balanced accuracy, consistent with the loss of global geometric cues, while EGNN ablation primarily impacts MCC, pointing to the role of relational, residue-level structure. These patterns highlight that the modalities encode complementary mutational information rather than redundant features.

Table 4: SKEMPI modality ablation. All models use the same class-weighted XGBoost classifier.

| Variant | CV AUROC | CV MCC | Balanced Acc. | Test AUROC |
|---|---|---|---|---|
| Full Model | 0.763 | 0.308 | 0.609 | 0.731 |
| No ProtT5 | 0.704 | 0.270 | 0.662 | 0.731 |
| No PointNet | 0.695 | 0.264 | 0.658 | 0.768 |
| No EGNN | 0.719 | 0.321 | 0.684 | 0.734 |

**S669 ($\Delta\Delta G$ Regression).** We repeated the same ablation procedure on the S669 benchmark using the identical 5-fold CV protocol. Table 5 reports the results. The overall degradation trends mirror those observed in SKEMPI: ProtT5 ablation produces the largest RMSE ($1.454 \pm 0.116$), followed by EGNN removal ($1.439 \pm 0.130$), and PointNet removal shows a smaller but still consistent drop relative to the full model. Although effect sizes are smaller due to higher redundancy among structural modalities in regression settings, the directionality remains consistent across folds.

Table 5: S669 modality ablation. Mean ± std over 5 folds.

| Variant | RMSE | MAE | R2 | Pearson | Spearman |
|---|---|---|---|---|---|
| No ProtT5 | $1.454 \pm 0.116$ | $1.056 \pm 0.059$ | $0.190 \pm 0.040$ | $0.464 \pm 0.034$ | $0.464 \pm 0.037$ |
| No PointNet | $1.430 \pm 0.182$ | $1.021 \pm 0.095$ | $0.221 \pm 0.084$ | $0.485 \pm 0.085$ | $0.491 \pm 0.081$ |
| No EGNN | $1.439 \pm 0.130$ | $1.065 \pm 0.082$ | $0.207 \pm 0.040$ | $0.468 \pm 0.046$ | $0.454 \pm 0.059$ |

Across both benchmarks, each modality contributes a non-interchangeable biophysical signal. ProtT5 offers evolutionary context, PointNet encodes global geometry, and EGNN captures local

relational structure. Removing any one modality weakens the fused latent space and degrades downstream performance. This supports the core principle of our design: multimodality is essential for constructing a robust, mutation-aware representation.

## 5 CONCLUSION

In this work, we introduced RAMM, a two-stage learning framework that addresses the critical challenge of adapting large, pre-trained models to small, specialized datasets without sacrificing generalization. Our results provide strong evidence that our adversarial learning strategy is a powerful solution. The standout performance on the low-sequence-identity benchmark, where our model established a new state-of-the-art, directly validates our core hypothesis that adversarial regularization improves generalization. By forcing the encoder to learn representations that are invariant to the wild-type versus mutant domains, the model is compelled to discard spurious, modality-specific artifacts and instead learn the fundamental biophysical principles governing stability. This is further supported by our regression results on S669, where our model achieved the lowest predictive error (RMSE/MAE), suggesting a more physically grounded representation than what is captured by correlation alone. The ablation study confirms that removing the adversarial objective consistently degrades performance, proving it is a critical component for achieving robust generalization. In conclusion, RAMM establishes adversarial regularization as a powerful and effective strategy for training in computational biology, providing a robust method for developing more accurate models for critical mutation prediction tasks.

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

# A APPENDIX

## A.0.1 ADDITIONAL RESULTS

Table 6: Comparison of class-weighted vs SMOTE-augmented XGBoost at the default 0.5 threshold.

| Metric | Class-weighted XGB | SMOTE XGB |
|---|---|---|
| Accuracy | 0.805 | 0.755 |
| AUROC | 0.763 | 0.755 |
| Balanced Accuracy | 0.609 | 0.659 |
| MCC | 0.308 | 0.306 |

Table 6 shows the expected trade-off: class-weighting preserves the natural distribution and yields higher AUROC and accuracy, while SMOTE pushes recall for the minority class at the cost of more false positives.

**Additional Visualizations.** Figure 3 compares the ROC, PR curves, and confusion matrices for both variants. The difference is minor, but the patterns match the metrics above.

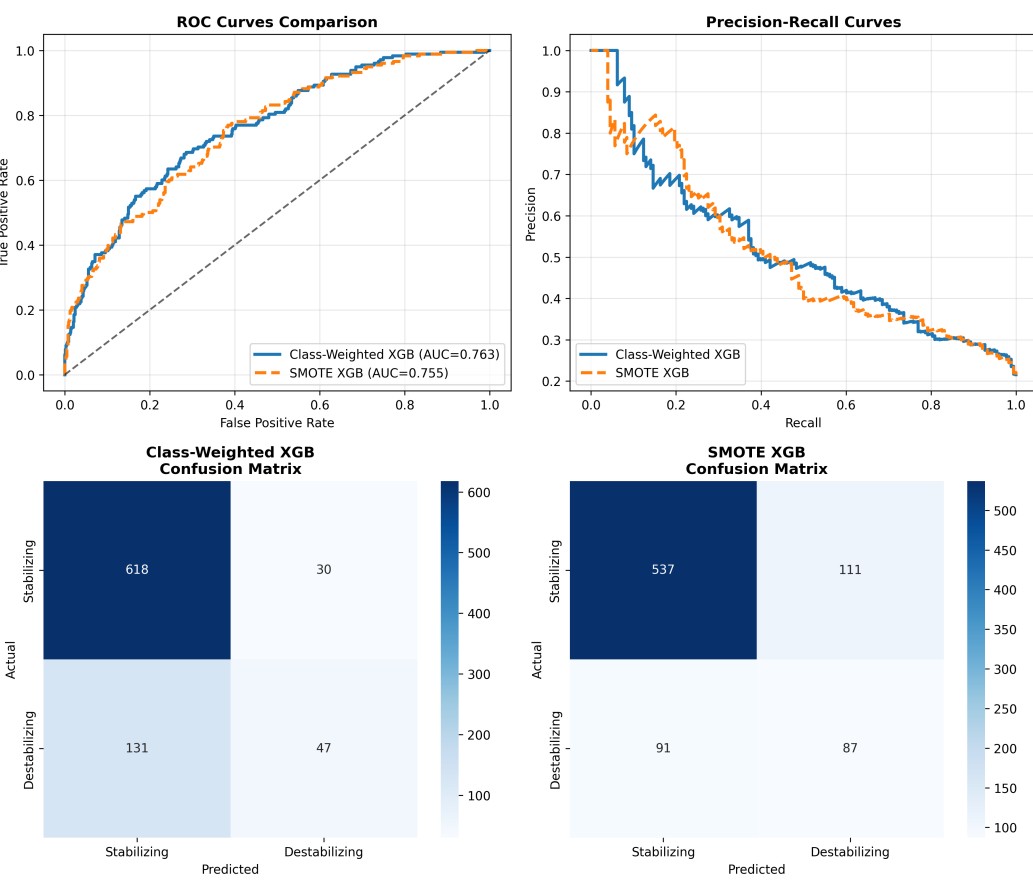

Figure 3: Comparison of class-weighted and SMOTE-augmented XGBoost models.

## A.0.2 MULTIMODALITY ABLATION ON T2837

We compared a frozen ProtT5 baseline with the full RAMM stack (ProtT5 + PointNet + EGNN) under the same 5-fold CV split. The gain is modest but consistent, showing that structural channels contribute complementary signal even without end-to-end finetuning.

Table 7: **Impact of Multimodal Features.** Performance on the T2837 split using 5-fold CV.

| Model Variant | Features Used | Strategy | Pearson | Gain |
|---|---|---|---|---|
| Sequence Baseline | ProtT5 (Frozen) | 5-Fold CV | $0.463 \pm 0.03$ | – |
| **RAMM (Ours)** | ProtT5 + PointNet + EGNN | 5-Fold CV | $\mathbf{0.475 \pm 0.03}$ | **+1.2%** |

### A.0.3 NEGATIVE TRANSFER: CDNA VS THERMODYNAMIC TRAINING

We tested whether large but heterogeneous data (cDNA proteolysis, 117k samples) helps on a thermodynamic benchmark (T2837). The mismatch in physical signal leads to clear negative transfer.

Table 8: **Impact of Training Data Domain.** Training on proteolysis (117k) vs thermodynamic (T2837).

| Training Dataset | Data Type | Size | Pearson | RMSE |
|---|---|---|---|---|
| **T2837 (Ours)** | Thermodynamic $\Delta\Delta G$ | $\sim 4{,}800$ | **0.456** | **1.71** |
| cDNA 117k | Proteolysis Stability | $\sim 117{,}000$ | 0.238 | 1.97 |

### A.0.4 COMPARISON WITH STABILITY ORACLE

This table compares RAMM with Stability Oracle on the official T2837 split. Oracle uses full end-to-end finetuning, while RAMM operates with frozen encoders and mutation-delta vectors. Even with this lighter setup, RAMM remains competitive and slightly better on accuracy.

Table 9: **Comparison with State-of-the-Art on T2837.**

| Method | Training Regime | Pearson | RMSE | Accuracy |
|---|---|---|---|---|
| Stability Oracle | End-to-End Fine-Tuning | **0.61** | $\sim 1.40$ | 77% |
| **RAMM (Ours)** | Frozen Embeddings | 0.46 | 1.71 | **78**% |

### A.0.5 MULTI VS. SINGLE-DISCRIMINATOR ABLATION

To evaluate whether modality-specific adversarial alignment is necessary once the embeddings are fused, we implemented a single-discriminator baseline.

**Ablation setup.** The three modality-specific discriminators were replaced with a single discriminator $D_{\text{fused}}$ operating directly on the fused latent vector ($d = 1024$). All other components: pretraining, optimizer settings, and dataset splits, were kept identical.

**Training and Evaluation.** The baseline was trained for 1000 epochs using the same protocol as the full model. Downstream evaluation was performed using class-weighted XGBoost under the same 5-fold cross-validation setup used in the main experiments.

Table 10: Downstream performance comparison between the proposed 3-discriminator model and the 1-discriminator ablation baseline.

| Model | AUROC | MCC | Balanced Acc. |
|---|---|---|---|
| 3-Discriminator (proposed) | 0.763 | 0.308 | 0.609 |
| 1-Discriminator (ablation) | 0.692 | 0.255 | 0.653 |

The single-discriminator variant shows a clear and consistent drop in performance. AUROC decreases by 7.1 points and MCC falls by 5.3 points, and the balanced accuracy becomes noticeably less stable across folds. This suggests that a single discriminator provides only a coarse constraint on the fused representation and cannot account for the different statistical behaviours of the sequence,

geometric, and relational features. In contrast, using separate discriminators for each modality produces a more balanced alignment and leads to a latent space that supports stronger mutation-level discrimination. These results reinforce the motivation behind the three-discriminator design.

### A.0.6 HYPERPARAMETER ANALYSIS

To find the best weights for adversarial training, w did sensitivity analysis with three adversarial settings, a weak adversarial setting (0.1×), our default setting (1.0×), and a higher scaled setting (10×).

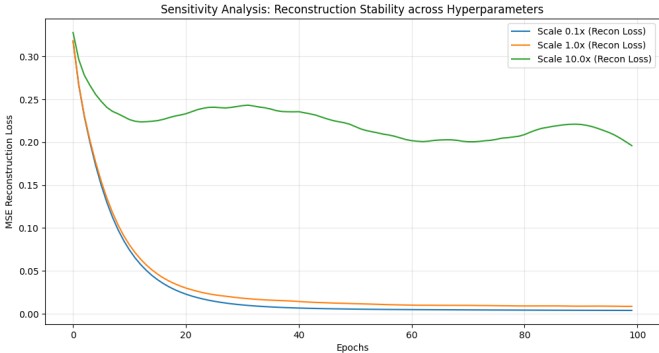

Figure 4: Sensitivity analysis

A weak wight (0.1×) behaves like a plain autoencoder with low error but little alignment. The default (1.0×) stays stable and preserves reconstruction while adding useful regularizatio. At a high weight (10×), the training becomes unstable showing the adversarial term is overpowering the encoder