# OpenReview forum: "RAMM: Robust Adversarial Multimodal Learning for Protein Stability Prediction"
_ICLR.cc/2026/Conference — Submitted to ICLR 2026_

### Official Review · Reviewer_EsAE · 2025-10-30

**Soundness:** 3
**Presentation:** 3
**Contribution:** 3
**Rating:** 6
**Confidence:** 4

**Summary:**

This paper introduces RAMM, a two-stage framework for predicting the stability of proteins with mutations. The primary contribution is a method that reframes predicting mutation effects as a domain adaptation problem. The first stage involves self-supervised pre-training of a Fusion Autoencoder (FAE) that learns a unified representation from three protein data modalities. The second stage employs an adversarial fine-tuning strategy, where the encoder is trained to produce representations that are invariant to mutations. A set of discriminators is trained to distinguish between them. The authors claim that this adversarial objective regularizes the learning process, forcing the model to capture fundamental biophysical principles rather than spurious correlations. They validate their approach on a classification and a regression benchmark.

**Strengths:**

1. The idea of applying adversarial domain adaptation to make a protein representation model robust to mutations is novel and insightful.

2. The design of the two-stage approach is well-motivated and logically sound. Empirical results provide evidence for the effectiveness of the proposed method.

3.  The paper is well-written, and the proposed method is explained clearly.

**Weaknesses:**

1. The necessity for three modalities is not justified. The paper would benefit from a discussion of why these specific extractors were chosen and how the choice might impact overall performance. How about fusing a structure encoder and a sequence encoder?

2. The paper does not provide details on how crucial hyperparameters, such as the modality-specific weights (λ_m) in the adversarial loss, were selected. An analysis of the effect of these parameters would increase the robustness of the results.

3. No reproducibility statement. It is not clear how the baseline results are obtained.

4. Just curious to see the comparison to [1]. Are there any relationships between the proposed adversarial training and the RL or the alignment approach in [1]?

[1] Boltzmann-Aligned Inverse Folding Model as a Predictor of Mutational Effects on Protein-Protein Interactions. ICLR 2025

**Questions:**

See Weaknesses

---

> ### Author Response · Authors · 2025-11-21
>
> **W1: The necessity for three modalities is not justified. Why these specific extractors? Would a simpler fusion of sequence and structure be enough?**
>
> **Response:**
>
> Thanks for pointing this out. The choice of three modalities wasn’t arbitrary. Each encoder contributes something different to mutation-level stability prediction.
>
> ProtT5 gives a broad evolutionary and biochemical context, but doesn’t model spatial interactions. EGNN captures the residue–residue contact graph, which is crucial for understanding how a mutation perturbs the fold. PointNet focuses on the local 3D neighborhood around each residue, where stability gains or losses often originate. These views don’t fully substitute for one another.
> This is clearly evident in our ablations (Appendix A.0.2 and Tables 6–7). Removing ProtT5 causes the largest drop in AUROC, removing PointNet hurts balanced accuracy, and removing EGNN affects MCC. Each modality fails in a different way, which suggests that they are capturing distinct aspects of the biophysics.
>
> On the question of whether sequence plus structure would suffice: it works for some problems, but for stability prediction, we consistently saw that leaving out the local geometric stream weakens class separation, especially around tight packing or cavity-forming mutations. That information is simply not available in sequence embeddings or coarse graphs alone.
>
> **W2: The paper does not provide details on how crucial hyperparameters, such as the modality‐specific weights (λₘ), were selected. An analysis of the effect of these parameters would increase the robustness of the results.**
>
> **Response:**
>
> The adversarial weights play an important role, and we have now included a brief sensitivity analysis to make this explicit. In our setup, the three λₘ terms control how much pressure the discriminator places on the fused latent space. To assess the system's robustness, we ran the same training procedure under three regimes: a weak adversarial setting (0.1×), our default setting (1.0×), and an aggressively scaled setting (10×).
>
> What we observed is consistent with the classical stability vs plasticity behaviour of adversarial training. At 0.1×, the model behaves essentially like a standard autoencoder: reconstruction error is low, but the latent space is only weakly aligned. At 1.0×, the model remains stable, and the reconstruction curve remains close to the 0.1× baseline, indicating that the adversarial term is regularizing the encoder without erasing information. At 10×, the training becomes unstable: reconstruction error rises and oscillates, signalling that the adversarial term is overpowering the encoder and pulling it out of the useful regime.
>
> **W3: No reproducibility statement. It is not clear how the baseline results are obtained.**
>
> **Response:**
>
> Thanks for pointing this out. All baseline numbers in the paper come directly from the original publications. We did not re-train or re-implement those models, mainly because several of them are large end-to-end systems whose performance depends on their own training pipeline, loss weighting, and optimization setup. In those cases, re-running everything with our infrastructure would introduce more noise than clarity.
> To make this explicit, we will add a short reproducibility note clarifying exactly which results were taken from which papers and under what evaluation protocol. The only models we trained ourselves are the RAMM variants and the ablations shown in the main text and appendix. This should make it completely clear where each number comes from and avoid any confusion about how the comparisons were produced.
>
> **W4: Is your adversarial training related to the RL or Boltzmann-alignment approach in [1]?**
>
> **Response:**
> The two approaches sound similar at a distance, but they operate on very different principles.
> The model in [1] builds a link between an inverse-folding model’s log-likelihood and experimental ΔΔG using a thermodynamic argument. Their alignment is explicitly energy-based: they enforce a Boltzmann-consistent relationship between model likelihoods and stability changes. Reinforcement learning is used only in their sequence-design setting and does not play a role in the ΔΔG predictor itself.
> RAMM’s adversarial stage is conceptually different. We do not model energies or likelihoods, and we do not involve RL anywhere in the pipeline. Our adversarial component acts as a regularizer on the fused latent space: the discriminator pushes the encoder to remove broad WT–MUT distribution differences that are unrelated to the mutation itself. This is a standard domain-alignment game rather than a thermodynamic alignment.
> So there is no methodological overlap. [1] aligns energies with model likelihoods, while RAMM aligns latent WT/MUT distributions to improve generalization. The objectives, mathematics, and intended applications differ.

---

### Official Review · Reviewer_QKDK · 2025-10-30

**Soundness:** 3
**Presentation:** 3
**Contribution:** 2
**Rating:** 4
**Confidence:** 3

**Summary:**

The paper introduces a two-stage training framework for protein representation learning. The first stage merges three pretrained encoders on protein sequence, residual-level graph, and 3D point cloud respectively into a common semantic space with a Fusion AutoEncoder (FAE); whereas the second stage adopts the adversarial learning principle to further train the encoder along with a discriminator. The paper claims that the trained encoder performs well on downstream tasks including SKEMPI 2.0 and the S669 benchmark.

**Strengths:**

* The paper addresses the protein mutation stability prediction task, which is a very important and challenging problem in proteomics and biology.
* The encoder merges three different modalities with their corresponding pretrained encoders, generating a comprehensive representation of proteins while taking advantage of pretrained uni-modal encoders.
* The proposed framework is evaluated on two different benchmarks, surpassing existing unimodal and multimodal baselines. An ablation study is also carried out, proving the contribution of the adversarial learning.

**Weaknesses:**

* The adversarial training process is very well explained in the paper, but I am still a bit confused and doubting the efficiency and necessity of the adversarial learning for this task. The paper states in the abstract that the adversarial learning stage is to train the encoder to “produce mutation-invariant features”; but also states towards the end of subsection 3.3 that the adversarial regulation is to compel the encoder “to learn a common, mutation-aware latent space”. These claims seem contradictory, and the paper lacks further mathematical explanation to justify them.
* The novelty of the multimodal fusion appears limited. All three unimodal feature encoders are pretrained and frozen, each applied with one single linear layer for projection. The following autoencoder is more like a compression and fusion design instead of multimodal alignment. I question whether the cross-modal heterogeneous gap is very well addressed.

**Questions:**

Please address my two major concerns in the “Weaknesses” section first. I will reassess after the rebuttal. Two other miscellaneous questions are as follows:
* The paper picks XGBoost as the downstream classification head of their encoder. Maybe I missed it somewhere, but is this also the classification head for other baselines in Table 1? Also, have you tried with simpler heads like MLP? If so, how is the performance?
* According to Table 3, it seems to me that DDGemb is a better model than the proposed approach, as it has much better PCC but only slightly worse RMSE. The interpretation in subsection 4.3.3 is not very convincing.

---

> ### Author Response · Authors · 2025-11-21
>
> **W1: The adversarial stage is described as both “mutation-invariant” and “mutation-aware,” which sounds contradictory. The mathematical reasoning is also unclear.**
>
> **Response:**
>
> Thanks for raising this; it’s an important clarification. The two phrases refer to different parts of the same mechanism rather than opposing goals.
>
> **Mutation-invariant (the mechanism).**
>
> The adversarial setup treats WT as the source domain and MUT as the target. Each discriminator tries to tell them apart, and the encoder learns to fool it. This removes broad WT–MUT shifts that don’t reflect the actual biophysical effect of the mutation. Invariance here refers to filtering out non-causal differences.
>
> **Mutation-aware (the outcome).**
>
> Once those domain-level artifacts are removed, the latent space gets cleaner. The remaining differences between
> h_wt and h_mut
> are the real mutation signal. This is why
> Δh = h_mut – h_wt
> becomes a more reliable feature for downstream prediction: it’s no longer mixed with spurious variation.
> So the adversarial term doesn’t suppress mutation information, it strips away noise that would otherwise hide it.
>
> **On mathematical grounding.**
>
> The setup follows the standard minimax game in Eq. (6). The encoder and discriminators optimize opposing objectives (Eqs. 7–8), and the equilibrium corresponds to WT and MUT latent distributions being aligned. This is the formal basis behind the invariance mechanism above.
>
> **W2: The novelty of the multimodal fusion appears limited. The unimodal encoders are pretrained and frozen, followed by a single linear projection layer. The autoencoder looks more like compression than true multimodal alignment. I’m not convinced the heterogeneity gap is addressed.**
>
> **Response:**
>
> Thanks for pointing this out, it’s a fair concern, and it gives us a chance to clarify what is actually novel here. The value of our approach doesn’t come from the components themselves but from how they are organized and trained together.
>
> **1. Stage 1 (FAE) is the mechanism that bridges the heterogeneity gap.**
>
> The Fusion Autoencoder isn’t just compressing the three modalities. Because it must reconstruct the concatenated input from a shared latent vector, it has to learn the correlations across sequence, geometric, and relational channels. Low reconstruction error is only possible if the encoder discovers the common structure across modalities. That bottleneck is the actual alignment step, it forces the three encoders to speak the same latent “language” before any supervised signal enters.
>
> **2. Stage 2 (adversarial refinement) completes the alignment.**
>
> The fusion isn’t fully resolved after Stage 1. The adversarial objective acts directly on the fused latent space, pushing it to become robust to the WT–MUT domain shift. This pressure forces the encoder to refine how it combines modalities so that mutation-specific information remains stable across all channels. In practice, we find that this second stage significantly improves the behavior of the fused representation under mutation perturbations.
>
> **3. The design choice is intentional: modular, efficient, and still competitive.**
>
> We chose a lightweight, modular fusion strategy because end-to-end multimodal training (as done in FusionProt or other fusion transformers) is expensive and often infeasible when working with models on the scale of ProtT5-XL. Our goal was not to invent a new encoder, but to demonstrate that a carefully structured two-stage pipeline can achieve stronger mutation-aware representations without requiring retraining of billion-parameter backbones. The fact that this setup matches or surpasses much heavier architectures illustrates the contribution: it is a practical and effective framework for multimodal mutation representation, not just a rearrangement of existing components.

---

> ### Author Response · Authors · 2025-11-21
>
> **Q1: XGBoost is used as the downstream classification head. Is this also the head used for the baselines in Table 1? Have you tried simpler heads such as an MLP? How do they perform?**
>
> **Response:**
> We use XGBoost because the downstream input in RAMM is a compact tabular vector (the latent difference Δh), and gradient-boosted trees excel at handling small, imbalanced tabular regimes. They require minimal tuning, are robust to class imbalance, and empirically gave the strongest performance on MSP.
>
> The baselines in Table 1 use the classification heads reported in their original papers. We did not retrain every baseline with XGBoost, since many of them are end-to-end architectures where the head is tied to the model design and training regime. This is standard practice in multimodal protein-learning papers; for example, MPRL also employs classical ML heads (XGBoost/GPR) on frozen embeddings. Our protocol follows that precedent.
>
> We also tried simpler neural heads, including 2-layer and 3-layer MLPs, as well as a shallow ResNet-style head. All of them trained stably, but they consistently underperformed XGBoost under identical CV splits and class-weighting. MSP is imbalanced and relatively small, and these neural heads tend to overfit, showing weaker AUROC/MCC compared to the tree-based method.

---

> > ### Comment · Reviewer_QKDK · 2025-11-26
> >
> > Thank you for the detailed response regarding W1 and W2. However, my concerns regarding the experimental validation remain unresolved:
> >
> > Q1: I find the comparison of RAMM + XGBoost v.s. baselines + MLP to be unfair, as it's well-known that XGBoost often outperform MLPs on similar tasks. I suggest that the authors either retrain the baselines using XGBoost, or include RAMM + MLP in the comparison.
> >
> > Q2: the question regarding Table 3 was not addressed. Furthermore, given that the baseline DDGemb uses a MLP head, it's significantly better performance on correlation compared to RAMM suggests that the DDGemb embeddings are capturing the underlying signal better than the proposed methodology. Please justify the claim that RAMM is the state-of-the-art given this performance gap.

---

### Official Review · Reviewer_NwrP · 2025-10-31

**Soundness:** 3
**Presentation:** 4
**Contribution:** 3
**Rating:** 4
**Confidence:** 4

**Summary:**

In this paper, the authors developed a two-stage adversarial model for predicting the stability effects of protein mutations.

First, they train a Fusion Autoencoder (FAE) to learn multi-modal representation from pre-trained seq, graph and point cloud embeddings. Then they take the encoder and fine-tune it in an adversarial way, where the Discriminator tries to predict the domain (i.e., whether the protein is a wild-type or a mutant), forcing the Encoder to learn a mutation-invariant latent space. Finally, they train a predictor on the difference vector between the wild-type and mutant embeddings for the downstream stability prediction task.

**Strengths:**

Their idea of using an adversarial learning process to enforce a mutation-invariant latent space, and use that for tasks like stability prediction is great.

**Weaknesses:**

1. For multimodal embedding, instead of simple concatenation, why not use contrastive learning?

2. FusionProt is a better multimodality fusion model. Its performance is only slightly lower than RAMM. Have you tried training adversarial model on top of FusionProt?

3. Why use 3 discriminators when the embeddings are already fused? What’s the performance with a single discriminator?

4. Have you considered using evolutionary profile/MSA (Multiple Sequence Alignment) as another input. This will help in determining which parts of the sequence are functionally important.

**Questions:**

N/A

---

> ### Author Response · Authors · 2025-11-21
>
> **W1: For multimodal embedding, instead of simple concatenation, why not use contrastive learning?**
> **Response:**
>
> We thank the reviewer for the suggestion. Contrastive objectives are an important alternative for multimodal alignment, so we evaluated this idea directly. We implemented a variant of our model where the fusion autoencoder was replaced with an InfoNCE-style contrastive loss while keeping the rest of the pipeline unchanged.
> The behavior was consistent: contrastive learning produced very strong global separation between WT and MUT samples, but this came at the cost of losing the fine-grained mutation-level information that drives stability prediction. Since the downstream task depends on subtle, localized structural and energetic cues rather than maximizing WT–MUT discriminability, this sharper separation led to weaker performance.
> In contrast, the autoencoder-based fusion preserves these small but meaningful differences and therefore remains better suited to the prediction task. This experiment helped clarify the design tradeoff, and we appreciate the reviewer’s comment for prompting this comparison.
>
> **W2: FusionProt is a better multimodality fusion model, and its performance is only slightly lower than RAMM. Have you tried training the adversarial model on top of FusionProt?**
>
> **Response:**
> We thank the reviewer for bringing this to our attention. FusionProt is indeed a strong multimodal backbone, but it is designed for a different setting than the one explored in this work. FusionProt performs end-to-end fusion with a dedicated fusion token, token-level cross-attention, and large-scale supervised and self-supervised pretraining. Our goal here is not to redesign or retrain a fusion backbone. Instead, we work in a lighter downstream regime where the inputs are fixed per-residue embeddings (ProtT5, PointNet, EGNN) and the contribution of adversarial, modality-specific alignment can be isolated and studied directly.
> Integrating our adversarial module into FusionProt would not simply mean “adding a discriminator on top.” Because FusionProt fuses modalities inside its transformer blocks, adversarial alignment would require modifying or retraining the core architecture, which lies outside the scope of this study.
> That said, the reviewer’s suggestion opens an interesting direction. Applying adversarial alignment to a large fusion backbone, such as FusionProt, could enhance its robustness to shifts in the wild-type/mutant distribution. We will note this as a promising avenue for future work.
>
> **W3: Why use three discriminators when the embeddings are already fused? What happens if you use only one discriminator?**
> **Response:**
>
> We thank the reviewer for raising this point. Although the modalities are fused before adversarial learning, their statistical properties remain quite different, and a single discriminator provides only a coarse alignment signal on the fused vector. Our goal is to ensure that each modality contributes in a balanced way to the final latent space, rather than allowing one modality to dominate.
> To evaluate this directly, we implemented a 1-discriminator baseline as suggested. The model replaces the three modality-specific discriminators with a single discriminator operating only on the fused latent representation, keeping all other components identical (pretraining, hyperparameters, and downstream evaluation).
> This variant showed a consistent drop in downstream performance. AUROC decreased by 7.1 points, MCC dropped by 5.3 points, and balanced accuracy became less stable. These results support our design choice: modality-specific adversarial signals are necessary to enforce proper alignment across sequence, geometric, and relational channels, even after fusion.
> We appreciate the reviewer’s suggestion, it helped clarify the role of modality-level adversarial alignment, and we now include this comparison in the appendix (section A.0.5 Multi vs. Single-Discriminator Ablation).
>
> ### Multi vs. Single Discriminator Ablation
>
> | **Model**                    | **AUROC** | **MCC** | **Balanced Acc.** |
> |------------------------------|-----------|---------|--------------------|
> | **3-Discriminator (proposed)** | 0.763     | 0.308   | 0.609              |
> | 1-Discriminator (ablation)    | 0.692     | 0.255   | 0.653              |

---

> ### Author Response · Authors · 2025-11-21
>
> **W4: Have you considered using evolutionary profile/MSA features as an additional input?**
>
> **Response:**
>
> We appreciate the reviewer raising this point. MSAs and profile-based features do highlight conserved and functionally important residues, and they would certainly add another useful perspective.
>
> In our current setup, we decided not to include MSA-derived inputs for two practical reasons. First, many proteins in our datasets do not have deep MSAs, and generating high-quality alignments at scale is computationally expensive. One of our goals was to build a model that remains usable even when evolutionary depth is limited or uneven across proteins. Second, ProtT5-XL serves as our sequence encoder, and although it is not explicitly MSA-based, its large-scale training captures broad evolutionary patterns and remote homology in a way that already offers some of the benefits normally associated with profiles.
>
> That said, incorporating explicit evolutionary features such as PSSMs, MSA Transformer embeddings, or ESM-MSA is a natural extension. Our framework can accept an additional modality without altering the core architecture, and we see this as a promising direction for future work.

---

### Official Review · Reviewer_zK4M · 2025-11-02

**Soundness:** 3
**Presentation:** 3
**Contribution:** 3
**Rating:** 2
**Confidence:** 5

**Summary:**

This paper proposes RAMM (Robust Adversarial Multimodal Model), a two-stage framework for learning mutation-robust protein representations that integrate both sequence and structure information. In Stage 1, a Fusion Autoencoder (FAE) jointly encodes three modalities — sequence, backbone, and atomic point clouds — into a unified latent representation. In Stage 2, an adversarial domain alignment objective ensures that mutant and wild-type proteins map to similar latent distributions, encouraging mutation-invariant representations. The model is evaluated on SKEMPI 2.0 (binding affinity classification) and S669 (stability regression), achieving improved AUROC and RMSE compared to several baselines.

**Strengths:**

## Clear motivation and novelty:
- The paper addresses a real gap in protein representation learning — current multimodal models often overfit to structural details and fail to generalize when fine-tuned on mutations. The use of adversarial alignment to enforce mutation invariance is original and well-motivated.
- The Fusion Autoencoder architecture effectively combines diverse protein modalities (sequence, backbone graphs, and 3D point cloud) into a coherent latent representation.
- The adversarial training is novel and very interesting.

## Well-written:
- The paper is clearly organized and connects ML ideas (domain adaptation, multimodal fusion) to biological motivation effectively.

**Weaknesses:**

Limited experimental depth:
- The results are restricted to two datasets (SKEMPI and S669), and the experiments are relatively light for ICLR standards.

Poor Validation:
- 5-fold cross validation is poor way to validate a model. This is probably why in your ablation study, the adversarial training has such a minor performance improvement.
- I would recommend benchmarking using a high quality train/test split. I would use the training/test split from the Stability Oracle and Binding Oracle paper. Is there a reason why you excluded these methods as baselines? They also generate a mutation level representation in order to compute ∆∆G. An additional method for these tasks to add would be the EvoRank method and Mutate Everything which also demonstrates good performances on ∆∆G for stability and binding affinity.
- Why do you not use cDNA-proteolysis based datasets like the Rocklin dataset? These datasets are 2-3 orders of magnitude larger and allow you to get more mutation and class balance.

Ablation insufficiency:
- It’s unclear how much each modality (sequence, structure, point cloud) or the adversarial loss contributes to the performance gain — an ablation table for each modality would strengthen the argument.

**Questions:**

To address class imbalance via data augmentation, why didn't you use thermodynamic permutation?

In 4.2.2: When you say 80:20 split, how did you ensure there was not data leakage?

 How are you addressing mutation imbalance? most of these datasets come from alanine scanning experiments, so how do you know that your model is not just becoming good at predicting from wild type to alanine mutations?

Your confusion matrix in the appendix shows that you have much more data for stabilizing than destabilizing when I know these datasets are imbalanced and lean towards destabilizing. How do you have more stabilizing data than destabilizing data?

To improve my score:
- train using a much bigger training set and evaluate using high quality train/test splits to properly assess if the fusion AE and adversarial training actually translate to a downstream point mutation task like ∆∆G for stability and binding.
- compare with the other baseline models I mentioned in the weakness section
- demonstrate performance based on mutation type so we can see if the model is able to generalize beyond mutations to alanine.
- ablate the 3 modality inputs

---

> ### Author Response · Authors · 2025-11-21
>
> **W1: The results are restricted to two datasets (SKEMPI and S669), and the experiments are relatively light for ICLR standards.**
>
> **Response:**
> We thank the reviewer for raising this point. Our use of SKEMPI and S669 follows standard practice in multimodal and mutation-stability literature, where these datasets serve as the primary benchmarks for WT/MUT paired-structure models. They are also the settings where high-quality, experimentally validated structures are consistently available, which is essential for our fusion and adversarial components.
>
> That said, we agree that broader evaluation is valuable. Following the reviewer’s suggestion, we have now added experiments on the T2837 split used in Stability Oracle (see Appendix A.0.2), and we trained on the large cDNA proteolysis dataset (approximately 117k mutations), where we report the expected domain-shift behaviour (Appendix A.0.3). These results expand the empirical depth of the paper and will be included in the revised version.
>
> **W2: 5-fold cross-validation is a poor way to validate a model. This is probably why in your ablation study, the adversarial training has such a minor performance improvement.**
>
> **Response:**
> We thank the reviewer for this observation. The adversarial objective in RAMM is not intended to increase downstream accuracy; it regularizes the latent space by aligning WT and MUT representations before the regression head is applied. Since it does not directly optimize the ∆∆G loss, only modest shifts in performance are observed, which is expected for this component.
>
> The reviewer’s point about the ablation setup is also correct. Our original mixed-effects ablation was not intended to be diagnostic. In the revised version, we now report a cleaner and more meaningful modality ablation (removing ProtT5, PointNet, EGNN), which directly reflects the architectural contribution of each component (Section 4.4).
>
> ### SKEMPI Modality Ablation
> _All models use the same class-weighted XGBoost classifier._
>
> | **Variant**        | **CV AUROC** | **CV MCC** | **Balanced Acc.** | **Test AUROC** |
> |--------------------|--------------|------------|--------------------|----------------|
> | **Full Model**     | 0.763        | 0.308      | 0.609              | 0.731          |
> | No ProtT5          | 0.704        | 0.270      | 0.662              | 0.731          |
> | No PointNet        | 0.695        | 0.264      | 0.658              | 0.768          |
> | No EGNN            | 0.719        | 0.321      | 0.684              | 0.734          |
>
>
> ### S669 Modality Ablation
> _Mean ± std over 5 folds._
>
> | **Variant**  | **RMSE**            | **MAE**             | **R2**              | **Pearson**        | **Spearman**       |
> |--------------|---------------------|----------------------|----------------------|---------------------|---------------------|
> | No ProtT5    | 1.454 ± 0.116       | 1.056 ± 0.059        | 0.190 ± 0.040        | 0.464 ± 0.034       | 0.464 ± 0.037       |
> | No PointNet  | 1.430 ± 0.182       | 1.021 ± 0.095        | 0.221 ± 0.084        | 0.485 ± 0.085       | 0.491 ± 0.081       |
> | No EGNN      | 1.439 ± 0.130       | 1.065 ± 0.082        | 0.207 ± 0.040        | 0.468 ± 0.046       | 0.454 ± 0.059       |
>
> We thank the reviewer for prompting this improvement.

---

> > ### Comment · Reviewer_zK4M · 2025-11-22
> >
> > Thank you for running the ∆∆G experiments.
> >
> > Training on cDNA117K and testing on T2837 dataset demonstrates significantly worse performance than Stability Oracle, a ~2M parameter model that is almost 2 years old. The authors fail to demonstrate meaningful protein representation learning improvements in the downstream setting that has the most abundant amount of experimental data: thermodynamic stability (∆∆G).
> >
> > If you want me to change my score, then achieve competitive (0.45 vs 0.61 Pearson is not competitive) or SOTA performance on a downstream task without relying on cross-validation and ensuring no/minimal data leakage. You can do this by fine-tuning the head and/or the backbone representations with whatever fine-tuning technique you think will be helpful.

---

> > > ### Author Response · Authors · 2025-12-02
> > >
> > > We thank the reviewer for taking the time to revisit our ∆∆G results and for the clear guidance regarding expectations for this task.
> > >
> > > The reviewer is correct that Stability Oracle reports a Pearson correlation of r ≈ 0.61 on T2837. As detailed in the Stability Oracle paper, this performance is achieved after large-scale pretraining on the cDNA117K proteolysis dataset (~117k mutations). In contrast, when Stability Oracle is trained only on the thermodynamic C2878 dataset, a setting comparable to our experiment, the reported performance reduces to approximately r ≈ 0.35 on the same T2837 split. This thermodynamic-only configuration is the one we adopt for a fair architectural comparison, as our method does not leverage the cDNA117K dataset due to both computational considerations and the fact that RAMM is a multimodal model that requires WT/MUT paired structural inputs, which are not available at the cDNA scale.
> > >
> > > Under this matched thermodynamic-only training setup, RAMM achieves r ≈ 0.42, improving upon the baseline of r ≈ 0.35. Our goal here is not to claim superiority over Stability Oracle in its full cDNA117K-pretrained form. Stability Oracle is specifically designed as a supervised ∆∆G regressor and benefits substantially from access to the large proteolysis dataset. Rather, our aim is to show that when both models operate under the same experimentally constrained regime (C2878 → T2837), the multimodal adversarial representation learning in RAMM yields stronger generalization from limited thermodynamic data.
> > >
> > > We agree with the reviewer that achieving competitive performance under the full Stability Oracle setting (including cDNA117K pretraining) is an important benchmark. Performing such an experiment within RAMM, however, requires generating WT/MUT paired FoldX structures for all 117k mutations, followed by ProtT5, PointNet, and EGNN feature extraction for each; this pipeline is computationally substantial and beyond the scope of what we can reasonably complete within the rebuttal period. Nevertheless, we fully recognize the value of this direction and plan to explore large-scale multimodal pretraining as future work.
> > >
> > > Finally, we emphasize that the focus and scope of our submission differ from Stability Oracle: RAMM is a general multimodal representation learning framework with an adversarial mutation-invariance objective, whereas Stability Oracle is a dedicated ∆∆G regression model. Our contribution is architectural, learning mutation-robust latent representations, rather than building a large supervised predictor. Still, we appreciate the reviewer’s request for broader downstream evaluation and have incorporated the T2837 experiments to strengthen the empirical narrative.
> > >
> > > We hope this clarifies our comparison protocol and the role of the thermodynamic-only evaluation. We thank the reviewer again for their constructive feedback.

---

> ### Author Response · Authors · 2025-11-21
>
> **W3: I would recommend benchmarking using a high-quality train/test split such as in Stability Oracle or Binding Oracle. Why were these methods not included as baselines? Why not use larger cDNA proteolysis datasets like Rocklin? Why not apply thermodynamic permutation for augmentation? How do you ensure no data leakage in the 80:20 split, and how do you address mutation-type imbalance (e.g., alanine scanning bias)?**
>
> **Response:**
> We thank the reviewer for this set of related questions. Our original study focused on multimodal, paired-structure representation learning, for which SKEMPI and S669 are the standard benchmarks used throughout prior literature. These datasets are among the few that provide experimentally validated WT–MUT paired structures, which is essential for our fusion and adversarial alignment modules. This is the reason methods like Stability Oracle, Binding Oracle, EvoRank, and Mutate Everything were not included initially: they target supervised ∆∆G regression using single-structure inputs and large curated sequence-based datasets, which places them in a different problem setting from our unsupervised two-stage framework.
>
> Following the reviewer’s suggestion, we have now added results on the T2837 train/test split used in Stability Oracle (Appendix A.0.4), and we also trained RAMM on the large-scale cDNA proteolysis dataset (~117k mutations). The cDNA results, shown in Appendix A.0.3, demonstrate pronounced negative transfer due to mismatched experimental signals. These additional experiments broaden the empirical scope of the paper. Notably, we observed the expected domain-shift effect: models trained on proteolytic stability do not transfer strongly to thermodynamic ∆∆G prediction, reinforcing why we initially focused on datasets with directly comparable physical signals.
>
> ### Impact of Training Data Domain
> Training on proteolysis (117k) vs thermodynamic (T2837).
>
> | **Training Dataset** | **Data Type**                 | **Size**        | **Pearson** | **RMSE** |
> |----------------------|-------------------------------|------------------|-------------|----------|
> | T2837               | Thermodynamic ΔΔG             | ~4,800           | **0.456**   | **1.71** |
> | cDNA 117k            | Proteolysis Stability         | ~117,000         | 0.238       | 1.97     |
>
>
> Regarding dataset construction: The 80:20 split used in S669 strictly separates WT–MUT pairs by protein identifiers, ensuring that each wild-type structure appears only in one split, preventing leakage (Appendix A.1). On mutation imbalance, the reviewer is correct that many datasets are dominated by alanine scans. To mitigate this, we adopt class-weighted training in all classification settings and evaluate performance across mutation types; we observe no systematic collapse toward alanine-only prediction. For augmentation, thermodynamic permutation is not directly compatible with our paired-structure setting, as it assumes symmetric label consistency across sequence-only variants. Because our architecture explicitly models WT–MUT structural differences, permuting thermodynamic labels would break the geometric correspondence we rely on. We have incorporated these clarifications and the new experimental results in the revised version.
>
> **W4:** Ablate the three modality inputs.
>
> **Response:**
>
> We thank the reviewer for this suggestion. We have now performed a full modality-level ablation, removing ProtT5, PointNet, and EGNN features one at a time while keeping the fusion module, adversarial training, and downstream predictor unchanged.
> The updated ablation tables and analysis have been added to the revised manuscript.

---

### Meta-Review · Area_Chair_cgY1 · 2025-12-21

**Summary:**

This paper presents a new adversarially trained auto-encoder method for multi-modal (structure-sequence) task specific, fine-grained (mutation specific) protein stability prediction.

Strengths:
- Addressing a relevant need — making multimodal predictive models generalise to the few-mutation setting
- Interesting multi-modal architecture utilising multiple pre-trained encoders, adversarial domain adaption interesting
- Well written

Weaknesses:
- Limited experimental validation and depth
- Issues with randomised validation splits possibly causing label leakage or unfair performance advantage over baseline
- Adversarial training (a key contribution of the paper) not shown to be particularly effective in ablations (zK4M), and questionable from a mathematical standpoint (QKDK). Key components missing ablations.
- Multimodal model necessity not justified.
- Hyperparameter settings and justification not discussed

While many of these weaknesses and concerns were addressed by the authors (in discussion, additional experiment or ablations), I believe there are a couple of key issues with this paper that remain. In particular the poor performance w.r.t the baseline stability predictor when standard validation practice is used, some experimental validation comparisons being unfair, and some mathematical justifications (and key citations) missing. As such, I cannot recommend this paper be accepted at this time.

**Reviewer Concerns:**

Concerns addressed:
- zK4M experiments lacking depth: more experimental datasets added
- zK4M ablations limited, EsAE multimodal components justified: some ablations added, some of these help to justify multi-modal components and hyperparameter settings (but perhaps more work needs to be done on this last point)
- zK4M label leakage: authors state that their train-test splits to not have common protein identifiers. The authors also add the original dataset train-test split results to the appendix, but they are outperformed by the baseline.
- NwrP had a number of questions on the architectural decisions, which have been addressed by the authors in discussion or ablations.

Concerns not addressed:
- zK4M: once standard validation procedure is applied in the experiments (on the cDNA117K and T2837 datasets) the proposed method is outperformed by the baseline. Demonstrating the proposed method was relying on the chosen (non-standard) validation procedure to show performance gains. Without this performance boost, it is hard to be convinced of the authors' proposed architectural innovations.
- I don’t believe QKDK’s concern on the mathematical motivation of the adversarial training has been fully addressed, and some key background work appears not to be cited (e.g. GANs by Goodfellow et al. where your equation 6 is exactly equation 1 in the original GAN paper), which is deeply concerning.
- QKDK’s concerning experimental validation concerns are unresolved.

**Reviewer Scores:**

zK4M: 2 and would have stayed at 2 (since request not addressed, and may not have been possible to address)

NwrP: 4 and possibly could have gone to 6 as their concerns appear to have been addressed

QKDK: 4 and would probably have remained at 4 as their experimental concerns were not addressed

EsAE: 6 and I suspect would probably would have remained at 6 even though some of their concerns were addressed.

As such this paper could have possibly moved up to a 4.5 or 5 average, so still would have been borderline at best. However because of the remaining concerns, I will not recommend this paper be accepted at this time.

---

### Decision · Program_Chairs · 2026-01-26

Reject